# Pre-intervention characteristics of the mosquito species in Benin in preparation for a randomized controlled trial assessing the efficacy of dual active-ingredient long-lasting insecticidal nets for controlling insecticide-resistant malaria vectors

**Boulais Yovogan**[1,2◎], **Arthur Sovi** [2,3,4◎] *, **Gil G. Padonou**[1,2], **Constantin J. Adoha**[1,2], **Bruno Akinro**[2], **Saïd Chitou**[2], **Manfred Accrombessi** [4], **Edouard Dangbénon**[2], **Hilaire Akpovi**[2], **Louisa A. Messenger** [4,6], **Razaki Ossè**[2,7], **Aurore Ogouyemi Hounto**[8,9], **Jackie Cook**[10], **Immo Kleinschmidt**[10,11,12], **Corine Ngufor**[2,4], **Mark Rowland**[4], **Natacha Protopopoff**[4], **Martin C. Akogbéto**[2]

**1** Faculté des Sciences et Techniques de l'Université d'Abomey-Calavi, Abomey-Calavi, Benin, **2** Centre de Recherche Entomologique de Cotonou, Cotonou, Benin, **3** Faculté d'Agronomie, Université de Parakou, Parakou, Benin, **4** Faculty of Infectious and Tropical Diseases, Disease Control Department, London School of Hygiene and Tropical Medicine, London, United Kingdom, **5** Institut de Recherche Clinique du Bénin, Abomey-Calavi, Benin, **6** American Society for Microbiology, Washington, DC, United States of America, **7** Ecole de Gestion et d'Exploitation des Systèmes d'Elevage, Université Nationale d'Agriculture, Kétou, Benin, **8** Programme Nationale de Lutte Contre Le Paludisme (PNLP), Cotonou, Benin, **9** Faculté des Sciences de la Santé, Université d'Abomey-Calavi, Cotonou, Benin, **10** Medical Research Council (MRC) Tropical International Statistics and Epidemiology Epidemiology Group, London School of Hygiene and Tropical Medicine, London, United Kingdom, **11** School of Pathology, Faculty of Health Sciences, University of Witwatersrand, Johannesburg, South Africa, **12** Southern African Development Community Malaria Elimination Eight Secretariat, Windhoek, Namibia

◎ These authors contributed equally to this work.
* arthur.sovi@lshtm.ac.uk

## Abstract

### Background

This study provides detailed characteristics of vector populations in preparation for a three-arm cluster randomized controlled trial (RCT) aiming to compare the community impact of dual active-ingredient (AI) long-lasting insecticidal nets (LLINs) that combine two novel insecticide classes–chlorfenapyr or pyriproxifen–with alpha-cypermethrin to improve the prevention of malaria transmitted by insecticide-resistant vectors compared to standard pyrethroid LLINs.

### Methods

The study was carried out in 60 villages across Cove, Zangnanando and Ouinhi districts, southern Benin. Mosquito collections were performed using human landing catches (HLCs). After morphological identification, a sub-sample of *Anopheles gambiae* s.l. were dissected for parity, analyzed by PCR for species and presence of L1014F *kdr* mutation and by

**Data Availability Statement:** All relevant data are within the manuscript and its Supporting Information files.

**Funding:** This research is supported by a grant to the London School of Hygiene and Tropical Medicine from UNITAID and Global Fund via the Innovative Vector Control Consortium (IVCC). This cluster-randomized clinical trial is part of a larger project, "The New Nets Project".

**Competing interests:** The authors declare that they have no competing interests.

**Abbreviations:** LLINs, Long lasting insecticidal nets; RCT, Randomized controlled trial; AI, Active ingredient; HLC, Human landing catches; PCR, Polymerase chain reaction; CSP, Circumsporozoite protein; HBR, Human biting rate; b/p/n, bite/person/night; MFOs, mixed function oxidases; ib/p/m, infected bite/person/night; PBO, Piperonyl butoxide; kdr, Knock down resistance; ITN, Insecticide treated nets; WHO, World Health Organization; PPF, Piriproxyfen; HLC, Human landing catch; ODK, Open Data Kit; GPS, Global Positioning System; IRS, Indoor residual spraying; HBR, Human biting rate; SR, Sporozoite rate; EIR, Entomological inoculation rate.

ELISA-CSP to identify *Plasmodium falciparum* sporozoite infection. WHO susceptibility tube tests were performed by exposing adult *An. gambiae* s.l., collected as larvae from each district, to 0.05% alphacypermethrin, 0.75% permethrin, 0.1% bendiocarb and 0.25% pirimiphos-methyl. Synergist assays were also conducted with exposure first to 4% PBO followed by alpha-cypermethrin.

## Results

*An. gambiae* s.l. (n = 10807) was the main malaria vector complex found followed by *Anopheles funestus* s.l. (n = 397) and *Anopheles nili* (n = 82). *An. gambiae* s.l. was comprised of *An. coluzzii* (53.9%) and *An. gambiae* s.s. (46.1%), both displaying a frequency of the L1014F *kdr* mutation >80%. Although more than 80% of people slept under standard LLIN, human biting rate (HBR) in *An. gambiae* s.l. was higher indoors [26.5 bite/person/night (95% CI: 25.2–27.9)] than outdoors [18.5 b/p/n (95% CI: 17.4–19.6)], as were the trends for sporozoite rate (SR) [2.9% (95% CI: 1.7–4.8) vs 1.8% (95% CI: 0.6–3.8)] and entomological inoculation rate (EIR) [21.6 infected bites/person/month (95% CI: 20.4–22.8) vs 5.4 (95% CI: 4.8–6.0)]. Parous rate was 81.6% (95%CI: 75.4–88.4). *An. gambiae* s.l. was resistant to alpha-cypermethrin and permethrin but, fully susceptible to bendiocarb and pirimiphos-methyl. PBO pre-exposure followed by alpha-cypermethrin treatment induced a higher 24 hours mortality compared to alphacypermethrin alone but not exceeding 40%.

## Conclusions

Despite a high usage of standard pyrethroid LLINs, the study area is characterized by intense malaria transmission. The main vectors *An. coluzzii* and *An. gambiae* s.s. were both highly resistant to pyrethroids and displayed multiple resistance mechanisms, L1014F *kdr* mutation and mixed function oxidases. These conditions of the study area make it an appropriate site to conduct the trial that aims to assess the effect of novel dual-AI LLINs on malaria transmitted by insecticide-resistant vectors.

## Background

Malaria remains a major public health issue in Benin, with prevalence of infection within the general population ranging from 11% to 51% depending on the region, with high burden in children under 5 years old [1]. Long-lasting insecticidal nets (LLINs) distributed at the national level every three years, and indoor residual spraying (IRS) in targeted districts, are the main pillars on which Benin's National Malaria Control Programme (NMCP) relies for protection against malaria vectors. During the most recent mass distribution campaign in 2017, pyrethroid-treated LLINs (Yorkool, PermaNet 2.0 and Dawa Plus 2.0), were widely distributed across the country. In 2018, LLIN usage was relatively high, with 71% of the national population reporting sleeping under a net, and 92% of households owning at least one LLIN [1].

According to Bhatt *et al.*, [2], malaria control interventions helped reduce malaria incidence by 40% in Africa between 2000 and 2015, with insecticide treated nets (ITNs) being the highest contributor (68% of cases averted). Similarly, reductions in malaria disease burden following the scale-up of LLINs have been routinely documented in trials conducted in several African countries including Kenya [3,4], and Benin [5,6]. Although LLINs are efficacious against

susceptible vector populations, more recent studies have demonstrated that LLINs performed below expectations in areas where vectors are resistant to insecticides used for nets treatments, notably pyrethroids [7,8].

Insecticide resistance has emerged and spread across Africa, including Benin [9]. Between 2016 and 2018, progress in reducing malaria cases worldwide had stalled [9], with one of the likely reasons being pervasive insecticide resistance among malaria vector populations.

Researchers and decision-makers now have high expectations of next generation LLINs which the World Health Organization (WHO) has encouraged manufacturers to develop for the control of resistant mosquitoes. Currently, next generation LLINs under evaluation include nets which combine a pyrethroid insecticide with either piperonyl butoxide (PBO; a synergist that inhibits mono-oxygenases implicated in resistance) or pyriproxyfen (PPF; a growth regulator that inhibits fecundity and fertility in insects) or chlorfenapyr (a pyrrole insecticide that disrupts mitochondrial oxidative phosphorylation). Cluster randomized controlled trials (RCTs) conducted in Tanzania [8], Uganda [10], and Burkina Faso [11] demonstrated that pyrethroid-PBO LLINs and pyrethroid-PPF LLINs provide more protection against malaria than standard LLINs. Other insecticide mixture LLINs combining alpha-cypermethrin and chlorfenapyr [12] or pyriproxyfen [13], have showed superior efficacy compared to standard LLINs on entomological outcomes in experimental hut trials. However, there remains a dearth of evidence regarding the effectiveness of the latter two dual-AI LLINs on malaria infection and transmission at the community-level.

This study conducted in southern Benin presents baseline entomological data collected in preparation for an RCT assessing the effectiveness, of Interceptor G2 (a pyrethroid-chlorfenapyr LLIN) and Royal Guard (a pyrethroid-pyriproxifen LLIN) deployed in the community, on malaria incidence, prevalence and transmission.

## Material and methods

### Study area

The study was carried out in three adjacent districts [Covè (07˚13'08.0400" N, 02˚20'21.8400" E), Ouinhi (07˚05′00″ N, 02˚29′00″ E) and, Zagnanando (07˚16′00″ N, 02˚21′00″ E)] located 150 kilometers away from Cotonou, the economic capital of Benin. The area has two rainy seasons (May-July and September-November) and was selected because of its high endemicity for malaria, with infection prevalence of 36.5% in children aged under 5 years old [1], intense pyrethroid resistance in the primary malaria vector species [14] and proximity to experimental huts sites. The main economic activities carried out by the population are agriculture, fishing, hunting, trade, and hospitality industry. The main crops produced are groundnuts, rice, maize, oranges, cassava, beans, oil palm, sorghum and millet [15]. According to the study census performed in 2019, the area population was approximately 220,000. A total of 60 clusters were formed from the 123 villages in the study area. Entomological monitoring was conducted in one village in each cluster, equating to a total of 60 villages with 8 in Cove district, 33 in Zagnanando and 19 in Ouinhi (Fig 1).

### Human landing catches (HLC)

**Mosquito sampling technique.** One round of mosquito collections was undertaken across villages, between September and October 2019. In each village, four houses were selected for mosquito sampling using HLCs. To facilitate the supervision of mosquito collectors for HLCs, the first house was randomly selected from the study census list, while the other three were chosen by the field team, within a radius of 15–20 meters around the first one. Four collectors were required per house. Two collectors (one indoor and one outdoor) collected

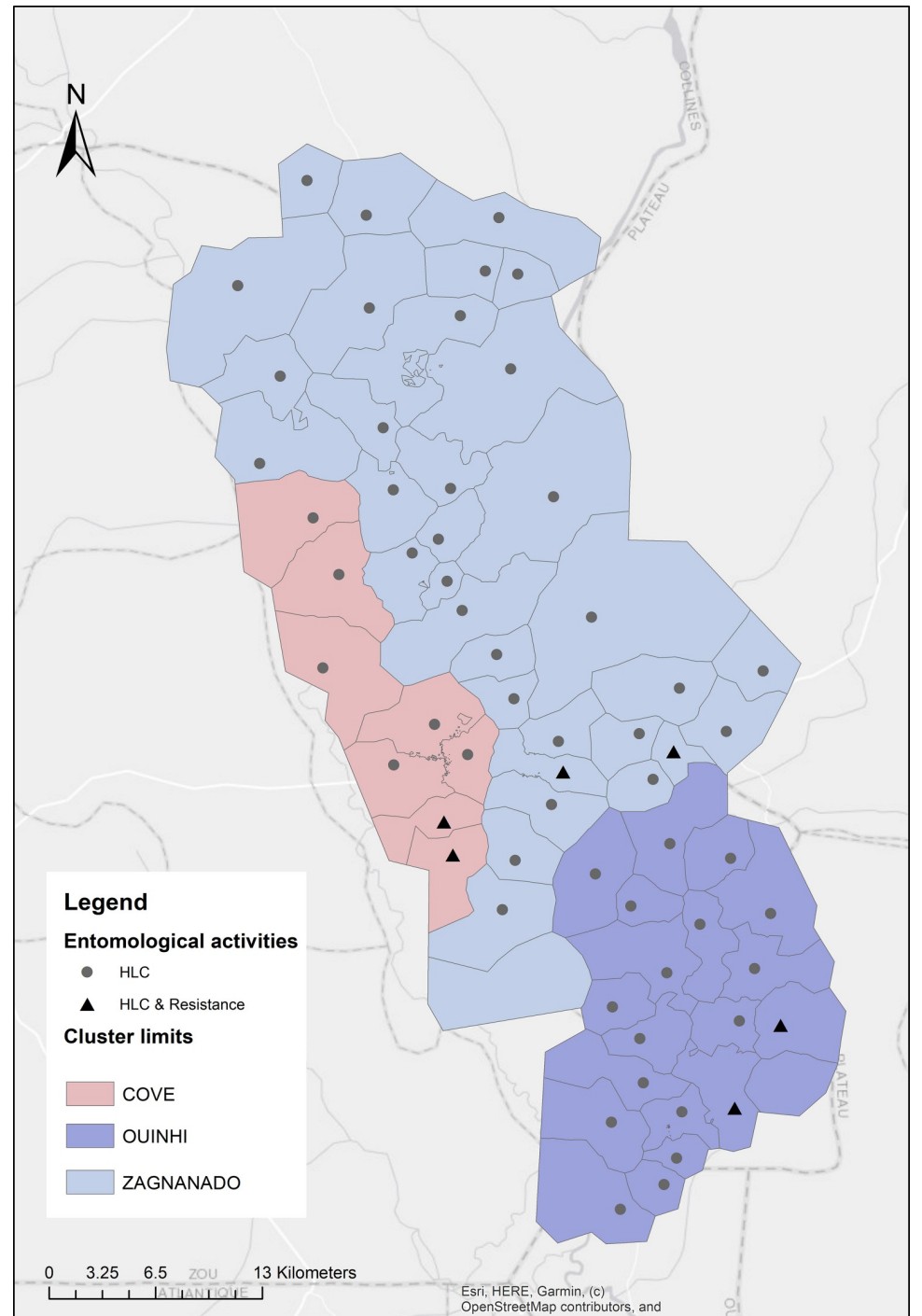

**Fig 1. Map of the study area.**

mosquitoes during 6 hours from 07:00 p.m. to 01:00 a.m. and the second group from 01:00 to 07:00 a.m. The collectors sat on a chair with their lower limbs exposed and collected all mosquitoes which landed on them using sucking tubes. To characterize *Anopheles* biting behaviour, collections were recorded per hour both indoors and outdoors.

**Mosquito processing.**    Mosquito specimens collected in HLCs were morphologically identified to species-level using the Gillies and Meillon [16] taxonomic key. In each village, a subsample of *An. gambiae* s.l. from indoor and outdoor collection and across collection hours were randomly selected, dissected to determine parous status [17], and analyzed by ELISA-CSP to detect presence of *Plasmodium falciparum*, following the protocol of Wirtz *et al.* [18]. Abdomens, legs and wings of *An. gambiae* s.l., previously analysed by ELISA-CSP, were used for species identification and L1014F *kdr* mutation following the protocols of Santolamazza et al. [19] and Martinez-Torres et al. [20], respectively.

**Household data collection.**    A short questionnaire about LLIN use was administered in houses where HLCs were conducted using Open Data Kit (ODK). The questionnaire recorded information about number of inhabitants in the surveyed households, number of people sleeping indoors and outdoors, number of people sleeping under nets, type of house (mud, cement, others), type of nets presents in the house, condition of nets, GPS coordinates of households and, other malaria prevention measures (IRS, coils, any others) used by household members.

## WHO susceptibility tube tests

**Mosquito larvae collections and rearing.**    Mosquito larvae and pupae were sampled from various breeding sites in 2 nearby villages in each district, using a larval dipper. They were transported to the field insectary for rearing until adulthood at 25°C ± 2°C and 80% ± 10% relative humidity. After emergence, morphological identification of adult mosquitoes was performed to species-level, and only *An. gambiae* s.l. individuals were tested.

**Susceptibility testing.**    In each district, batches of 20–25 unfed females *An. gambiae* s.l. aged 3–5 days old were aspirated into four tubes containing WHO insecticide impregnated papers (0.75% permethrin, 0.05% alpha-cypermethrin, 0.1% bendiocarb or 0.25% pirimiphosmethyl) for one hour. Separate batches were exposed to a tube lined with a WHO control paper in parallel.

To evaluate the involvement of mixed function oxidases (MFOs) in pyrethroid resistance in populations of *An. gambiae* s.l., synergist assays with piperonyl butoxide (PBO; 4%) were performed. Mosquitoes were pre-exposed to PBO papers in WHO tubes for one hour, before transfer to different tubes with alpha-cypermethrin papers (0.05%) for a further hour.

The percentage mortality at 24, 48, and 72 hours post-exposure was recorded. All tests were performed following the WHO protocol [21].

## Ethical considerations

The protocol of the present study has been reviewed and approved by the Benin national ethics committee for health research (N˚30/MS/DC/SGM/DRFMT/CNERS/SA, Approval n˚6 of 04/03/2019) and the ethics committee at the London School of Hygiene and Tropical Medicine (16237–1). Written consent to participate in the study was taken from head of households and adult volunteers who performed HLCs after they have been fully informed of the risks of the study. Collectors were trained to collect any mosquito that landed on them before being bitten. All fieldworkers have been vaccinated against yellow fever. When they experienced malaria symptoms, they were immediately provided with anti-malarial medication such as Artemisinin-based Combination Therapy in the nearest health facility.

## Data management and analysis

Entomological monitoring data were double entered into databases designed in CS Pro 7.2 software and, cleaned with Stata 15.0 (Stata Corp., College Station, TX).

Entomological indicators of malaria transmission measured both indoors and outdoors, were determined as mentioned in the Table 1 below.

As all *An. gambiae* s.l. positive for CSP ELISA and approximately 50% of the negative ones were tested for molecular species identification, the SR per molecular species (*An. coluzzii* and *An. gambiae* s.s.) was weighted to account for proportion of collected *Anopheles* processed for CSP. This allowed taking into account the unequal sampling.

The mean of the household results for HBR, SR and EIR were used to generate a village level result. The mean of the village results is presented by district and their confidence intervals were calculated using the Poisson distribution.

According to the WHO guidelines [21], resistance status of populations of *An. gambiae* s.l. was determined after exposure to the discriminating insecticide dose, as follows:

i.  Susceptible (mortality rate ≥ 98%)

ii. Possible resistance (mortality rate between 90–97%)

iii. Resistance (mortality <90%)

According to the WHO guidelines [21], involvement of metabolic mechanisms in insecticide resistance in populations of *An. gambiae* s.l. was determined as follows:

i.  Metabolic mechanism not involved (insecticide-synergist mortality not higher than for insecticide-only)

ii. Metabolic mechanism partially involved (insecticide-synergist <98% mortality but higher than for insecticide-only)

iii. Metabolic mechanism fully involved (insecticide-synergist ≥98% and higher than for insecticide-only)

Confidence intervals of mortality rates were determined using the exact binomial test. All statistical analyses were performed using Stata version 15.0 (Stata Corp., College Station, TX).

## Results

### Household and individual characteristics of study population

A total of 240 households were visited for HLCs, the average number of habitants per household was 4.5 (95% confidence interval (CI): 4.2–4.8) (Table 2). The majority (92.8%, 95% CI: 87.2–98.8) of the population slept indoors with no difference between districts. All visited households owned at least one LLIN. The proportion of people that report to sleep under nets the previous night was similar across districts with a mean of 82.7% (95% CI: 77.3–88.3) for the study area.

Overall, the majority of houses were made of mud (65.8%, 95% CI: 55.9–76.9) and cement (26.7%, 95% CI: 20.5–34.1). At the district level, while most houses were made of mud in Zangnanando and Ouinhi, mud and cement made houses were found in similar proportions in the

**Table 1. Formulas of entomological indicators of malaria transmission.**

| Indicators | Formulas |
|---|---|
| Human biting rate (HBR) | Total *An. gambiae* s.l./number of collector night |
| Sporozoite rate (SR) | Number of positive mosquitoes/Total number tested |
| Parous rate | Number of parous mosquitoes/Total number dissected |
| Monthly EIR | HBR x SR x 30 |

**Table 2. Household and individual characteristics of study population.**

| Indicators | Cove | Zangnanado | Ouinhi | Study area |
|---|---|---|---|---|
| | (95% CI), N | (95% CI), N | (95% CI), N | (95% CI), N |
| **Households (HH)** | | | | |
| Total N of people | 69960 | 73733 | 72596 | 216289 |
| Total N of HH | 16941 | 18470 | 18732 | 54143 |
| N of visited HH | 32 | 132 | 76 | 240 |
| Mean N of people per visited HH | 4.3 (3.5–5.1), 137 | 4.7 (4.3–5.1), 624 | 4.4 (3.8–4.9), 331 | 4.5 (4.2–4.8), 1092 |
| Proportion of visited HH with at least one LLIN | 100% | 100% | 100% | 100% |
| People sleeping indoors in the visited HH | 97.1% (81.2–100), 133 | 92.9% (85.5–100), 580 | 90.9% (80.9–100), 301 | 92.8% (87.2–98.8), 1014 |
| People sleeping under nets the previous night in the visited HH | 89.1% (73.9–100), 122 | 81.9% (74.9–89.3), 511 | 81.6% (72.1–91.9), 270 | 82.7% (77.3–88.3), 903 |
| **Type of housing** | | | | |
| Cement wall | 40.6% (21.6–69.5), 13 | 23.5% (15.9–33.3), 31 | 26.3% (16.0–40.6), 20 | 26.7% (20.5–34.1), 64 |
| Mud wall | 43.8% (23.9–73.4), 14 | 72% (58.2–87.9), 95 | 64.5% (47.6–85.2), 49 | 65.8% (55.9–76.9), 158 |
| Other type of wall | 15.6% (5.0–36.5), 5 | 4.5% (1.6–9.9), 6 | 9.2% (3.7–18.9), 7 | 7.5% (4.4–11.9), 18 |
| **LLINs** | | | | |
| Total number of LLINs | 62 | 237 | 148 | 447 |
| Mean N of LLINs per HH | 1.9 (1.4–2.5) | 1.8 (1.5–2.0) | 1.9 (1.6–2.3) | 1.9 (1.6–2.0) |
| Permanet 2.0 | 79% (58.4–100), 49 | 88.6% (77.0–100), 210 | 90.5 (75.8–100), 134 | 87.9% (79.4–97.1), 393 |
| Other LLINs | 21% (11.1–35.9), 13 | 11.4% (7.5–16.6), 27 | 9.5 (5.1–15.9), 14 | 12.1% (9.8–15.8), 54 |
| Holed nets | 45.2% (30–65.3), 28 | 43.9% (35.8–53.2), 104 | 51.4 (40–64.3), 76 | 46.5% (40.4–53.3), 208 |

N: Number of, HH: Household.

more urban district of Cove. The mean number of LLINs per household was 1.9 (95% CI: 1.6–2.0) in the study area and, similar across districts. The majority (87.9%, 95% CI: 79.4–97.1) of nets used were PermaNet 2.0. The other types of LLINs used were: OlysetNet, DawaPlus, DuraNet and Yorkool. On average, 46.5% (95% CI: 40.4–53.3) of LLINs had holes with no difference between districts (Table 2).

## Mosquito species composition

A total of 46,613 mosquitoes were collected with HLCs. *Culex* and *Mansonia* accounted for 35.3% and 36.9% of all the collection respectively, and their proportions were slightly higher outdoors than indoors. *An. gambiae* s.l. was the most abundant of the *Anopheles* species and represented 28.2% and 18.4% of the collection indoors and outdoors respectively. Other species found in lower density were *An. funestus*, *An. nili*, *An. ziemanni*, and *Aedes* spp. collected both indoors and outdoors. *Coquilletidia* spp. and *Eretmapodites* spp. were captured only indoors but at very low frequencies (<1%) (Fig 2).

Molecular species identification performed on 1797 specimens showed that 53.9% (N = 968) were *An. coluzzii* and the remaining were *An. gambiae* s.s. Indoors, both species were in similar proportions [50.9%, 95% CI: 48.0–53.9 for *An. coluzzii* vs 49.1%, 95% CI: 46.1–52.0 for *An. gambiae* s.s.]. Outdoors, *An. coluzzii* predominated (59.3%, 95% CI: 55.3–63.1) over *An. gambiae* s.s. (40.7%, 95% CI: 36.8–44.7).

## HBR, SR and monthly EIR in *An. gambiae* s.l.

Overall, the mean HBR was higher indoors with 26.5 bites per person per night (b/p/n) (95% CI: 25.2–27.9), n = 6373] compared to outdoors [18.5 b/p/n (95% CI: 17.4–19.6), n = 4434] in the study area. The same trend was observed in all three study districts (Table 3).

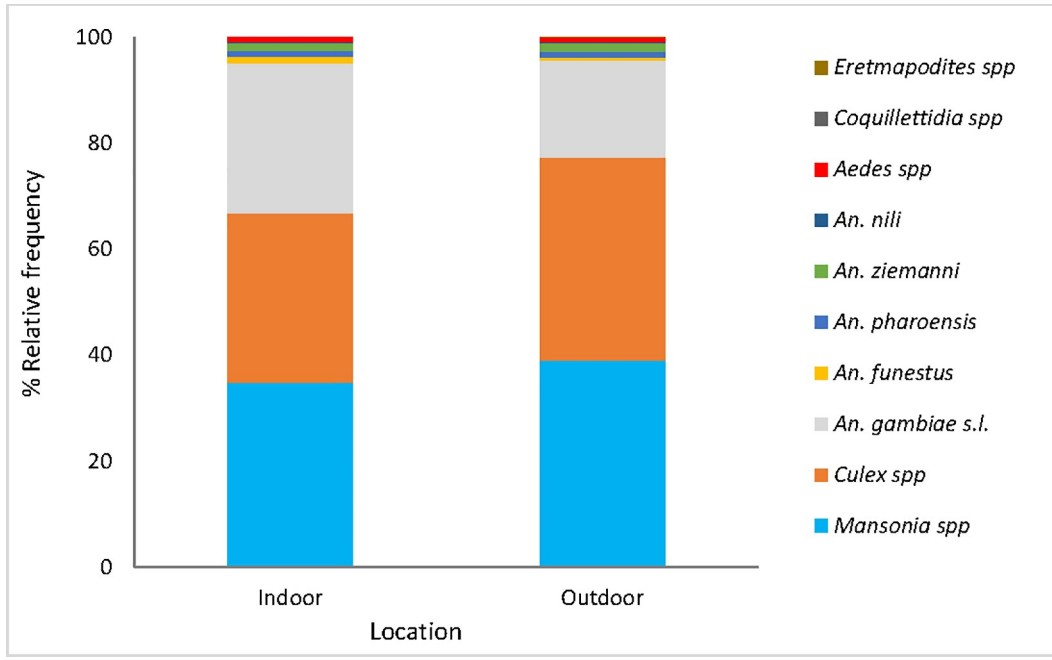

**Fig 2. Mosquito species composition in the study area.**

Biting of *An. gambiae* s.l. was more intense late at night both indoors and outdoors, with the lowest density before 10 pm. The peak in biting was 4.9 b/p/h (95% CI: 4.4–5.5) indoor and 3.3 b/p/h (95% CI: 2.9–3.8) outdoor and occurred between 4 and 6 a.m (Fig 3). The trend was the same at the study district level (Supporting information files, S1 Fig).

In *An. gambiae* s.l., the SR was higher indoors [2.9% (95% CI: 1.7–4.8), n = 2264] than outdoors [1.8% (95% CI: 0.6–3.8), n = 1341] in the study area with no significant difference, so was the trend in each district (Table 3).

At the molecular species level, the SR was 3.1% (95% CI: 2.3–4.0) in *An. gambiae* s.s. compared to 2.1% (95% CI: 1.5–2.8) in *An. coluzzii*.

Combined data revealed a higher monthly EIR indoors [21.6 b/p/m (95% CI: 20.4–22.8)] compared to outdoors [5.4 b/p/m (95% CI: 4.8–6.0)] in the study area, with the same trend in the three study districts (Table 3).

**Table 3. HBR, SR and EIR in *An. gambiae* s.l. in Cove, Ouinhi and Zangnanado.**

| Districts | Biting location | N of *An* collected | Person night | HBR (95%CI) | N of *An* tested | SR (95%CI) | EIR (95%CI) |
|---|---|---|---|---|---|---|---|
| Cove | Indoor | 975 | 32 | 30.5[a] (26.7–34.6) | 366 | 3.3[a] (0.6–9.5) | 21.4[a] (18.2–24.8) |
| | Outdoor | 556 | 32 | 17.4[b] (14.6–20.5) | 163 | 2.4[a] (0.06–13.4) | 6.4[b] (4.7–8.4) |
| Zangnanado | Indoor | 2792 | 132 | 21.2[a] (19.6–22.8) | 1187 | 3.7[a] (1.8–6.5) | 24.1[a] (22.4–25.9) |
| | Outdoor | 2134 | 132 | 16.2[b] (14.8–17.6) | 719 | 2.1[a] (0.5–5.4) | 5.9[b] (4.9–6.7) |
| Ouinhi | Indoor | 2606 | 76 | 34.3[a] (31.7–37.1) | 711 | 1.7[a] (0.3–4.9) | 17.3[a] (15.4–19.3) |
| | Outdoor | 1744 | 76 | 22.9[b] (20.8–25.2) | 459 | 1.7[a] (0.2–6.3) | 4.3[b] (3.4–5.4) |
| Study area | Indoor | 6373 | 240 | 26.5[a] (25.2–27.9) | 2264 | 2.9[a] (1.7–4.8) | 21.6[a] (20.4–22.8) |
| | Outdoor | 4434 | 240 | 18.5[b] (17.4–19.6) | 1341 | 1.8[a] (0.6–3.8) | 5.4[b] (4.8–6.0) |

An: *An. gambiae* s.l., The SR is expressed in percentage (%), The HBR was expressed in the number of bites/person/night (b/p/n), The EIR is expressed in number of infected bites/person/month (ib/p/m)

[a,b]: Indicator values with different superscripts within a same district are significantly different (p<0.05).

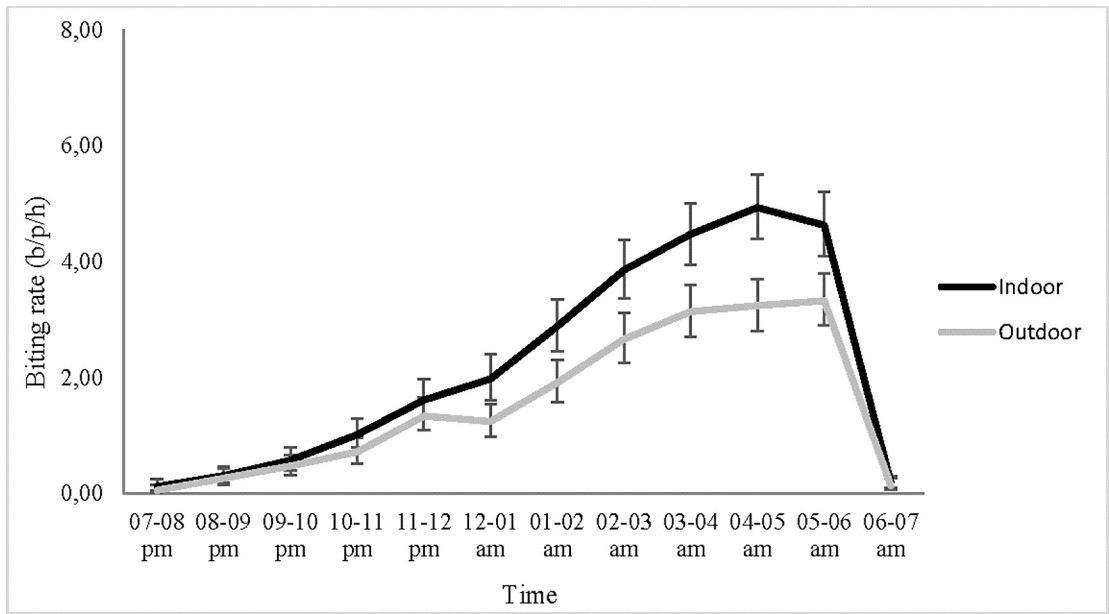

**Fig 3. *An. gambiae* s.l. hourly biting rates in the study area (N = 6373 indoors, N = 4434 outdoors), b/p/h: Bite/person/hour, the error bars indicate the confidence intervals.**

### Parous rate in *An. gambiae* s.l.

Of the 2,843 specimens of *An. gambiae* s.l. dissected in the study area, 2,327 were parous equating to a parous rate of 81.6% (95% CI: 75.4–88.4). Similar parous rates were observed indoors (82.2%, 95% CI: 74.0–91.0) and outdoors (80.7%, 95% CI: 70.7–91.8) in the study area. The same trend was observed in all three study districts (Supporting information files, S1 Table).

### WHO susceptibility tube tests and L1014F *kdr* mutation frequency

Susceptibility tube testing showed that *An. gambiae* s.l. populations were resistant to pyrethroid insecticides (alpha-cypermethrin and permethrin). By comparison, full susceptibility was observed to bendiocarb and pirimiphos-methyl. No significant difference was observed between the mortality rates at 24h, 48h and 72h, post-exposure (Fig 4) per district or per insecticide. While full susceptibility was not reached, pre-exposure to PBO increased mortality to alpha-cypermethrin from 9.6% (95% CI: 5.7–14.9), 4.5% (95% CI: 2–8.7), 8.6 (95% CI: 5.1–13.5) to 21.4% (95% CI: 15.7–28), 37.3% (95% CI: 30.5–44.5), 27.3% (95% CI: 17.7–38.6), respectively in Cove, Zangnanando and Ouinhi (Fig 5). This indicates partial involvement of mixed function oxidases (MFOs) in pyrethroid resistance in these *An. gambiae* s.l. populations.

Frequency of the L1014F *kdr* mutation in *An. gambiae* s.s. was 89.8% (95% CI: 88.2–91.2, n = 829) compared to 84.3% (95% CI:82.5–85.9, n = 968) in *An. coluzzii* collected during HLC. At the district level, the same trend was observed in Zangnanando, while the frequency of the mutation was similar in both species in Cove and Ouinhi (Supporting information files, S2 Table).

## Discussion

The current study provides detailed characteristics of vector populations in the study area in preparation for a cluster RCT which will assess the impact of community use of two dual-AI LLINs in the Zou region, Benin.

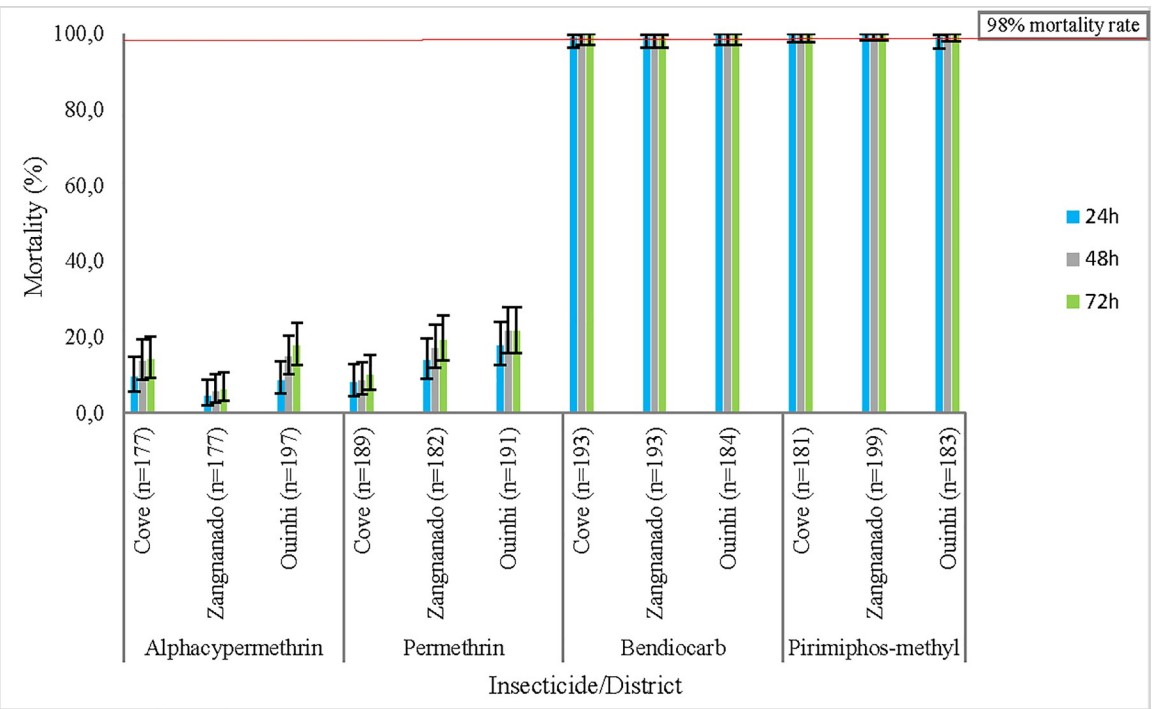

**Fig 4. Mortality rates of *An. gambiae* s.l. to 0.05% alpha-cypermethrin, 0.75% permethrin, 0.1% Bendiocarb and, 0.25% Pirimiphos-methyl, the error bars indicate the confidence intervals.**

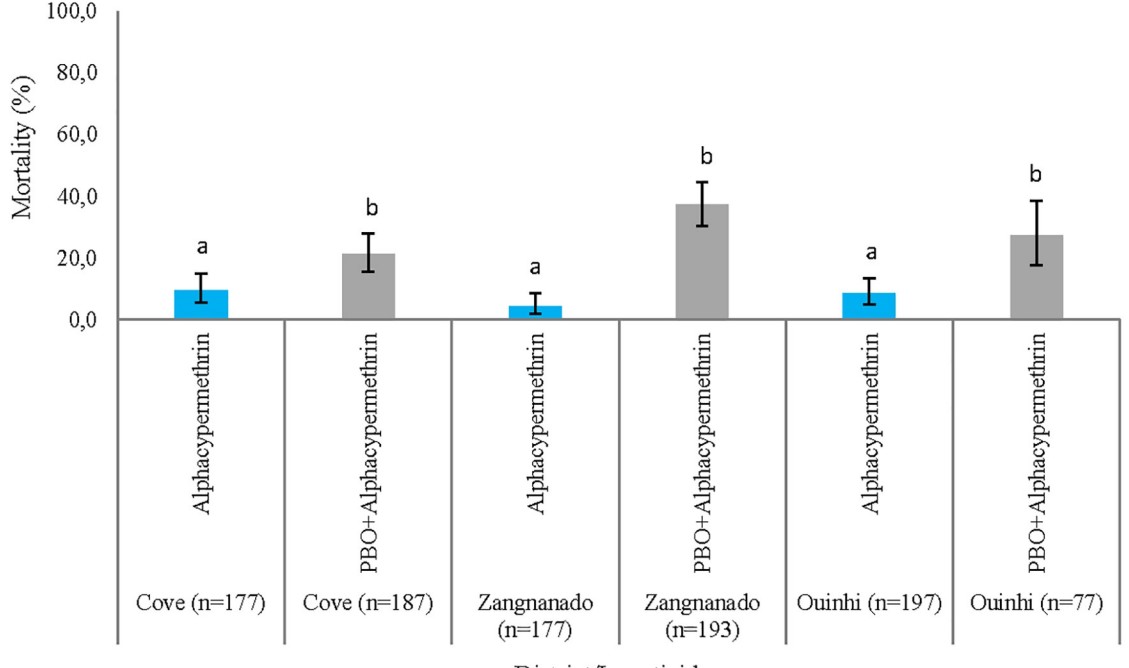

**Fig 5. 24 hours Mortality of *An. gambiae* s.l. to alpha-cypermethrin and PBO+alpha-cypermethrin in Cove, Ouinhi and Zangnanado, the error bars indicate the confidence intervals.**

Overall, *Anopheles gambiae* s.l. was the major vector complex, followed by *Anopheles funestus* and *Anopheles nili*, as previously reported in other regions of Benin [22–24]. Mosquito species from the *Mansonia* and *Culex* genera were also collected at moderate frequency (<40%), unlike *Aedes*, *Coquilletidia* and *Eretmapodites* which were found in low proportions (<2%). All collected mosquito species were found both indoors and outdoors except for *Coquilletidia* spp. and *Eretmapodites* spp. which were only collected indoors. The strong presence of mosquito species from the *Culex* and *Aedes* genera suggests a high potential for transmission of lymphatic filariasis and arbovirus such as dengue and yellow fever as observed in the Ouinhi [25] and Abomey-Calavi [26] districts. For that, apart from the LLINs whose distribution occurred as part of the RCT, a complementary strategy such as a larval source management program that will help discarding the majority of productive mosquito breeding sites would be needed.

Molecular species identification performed within the *An. gambiae* s.l. complex revealed the presence of a mixture of *An. coluzzii* (53.9%) and *An. gambiae* s.s. (46.1%). This is consistent with results from previous studies carried out in the same area by Ngufor et al. [14] and, in the neighbouring Plateau region by Sovi et al. [27]. According to Diabate et al. [28], permanent and semi-permanent breeding sites are conducive to the development of *An. coluzzii* while, temporary breeding site are favorable to the emergence of *An. gambiae* s.s. Indeed, several semi-permanent breeding sites have been created by the numerous tributaries of the Oueme and Zou rivers that water the study area as well as the presence of many rice growing areas. By comparison, temporary breeding sites were usually created by rain. The variation observed in the composition of the two molecular species between indoor and outdoor suggests that a scrutiny survey investigating the host-seeking behaviour of each species would be of interest. Despite the large number of specimens of *An. gambiae* s.l. analysed by PCR, *An. arabiensis* was not detected, although its presence in the neighbouring districts of Allada has been documented [29]. This could be due to the short mosquito sampling period and, the zoophilic behaviour of this species.

A large variability in the density of *An. gambiae* s.l. was observed among villages both indoors and outdoors in each district. This could be attributed to the disparity in the distribution of breeding sites from village to village. Despite the presence of LLINs in the majority of houses, and that more than 80% of household members slept under nets, the human biting rate in *An. gambiae* s.l was higher indoors than outdoors. This confirms the classical endophagic and anthropophagic behaviour which is observed in this mosquito species [30,31]. Indeed, this behaviour could have been facilitated by vector resistance to pyrethroids incorporated on the LLINs in use in the study area [14]. In addition, most biting was recorded late at night (from 11 p.m.), with peaks early in the early morning (around 4–5 a.m. or 5–6 a.m.). This suggests that the use of non-holed mosquito nets overnight can provide substantial protection to sleepers by significantly reducing the man-vector contact. According to a socio-anthropological study by N'tcha et al. [32], Benin's rural populations usually perform various nightly activities (children's play-activities, night talks, cooking, washing the dishes, eating, resting) outdoors and sleep under their nets indoors from 10 p.m. Thus, the little biting activity observed in *An. gambiae* s.l. early in the night (between 7-10pm) is not sufficient to put people at substantial elevated risk of receiving infected bites.

Populations of *An. gambiae* s.l. from all three study districts were resistant to pyrethroids but fully susceptible to bendiocarb and pirimiphos-methyl. A similar trend was observed by Gnanguenon et al. [33] and Sovi et al. [34] in the neighbouring districts of Allada and Pobe/Ketou, respectively. High use of LLINs over years, as well as the uncontrolled spray of insecticides for agricultural purposes throughout the region might have contributed to vector resistance to pyrethroids. While emergence of bendiocarb resistance occurred in some northern regions of the country where a carbamate-based IRS was implemented [35,36], continued

susceptibility to the same product and to pirimiphos-methyl was observed in the Cove, Ouinhi and, Zangnanando districts located in the south. This stresses the need for a judicious application of non-pyrethroid insecticides to delay the onset of resistance and preserve their efficacy.

The resistance genotyping revealed that the high frequency of L1014 *kdr* mutation could be partly responsible for pyrethroid resistance in *An. gambiae* s.l. Indeed, the frequency was high at 89.8% in *An. gambiae* s.s. and 84.3% in *An. coluzzii*. In addition, the synergistic assay data suggests the partial involvement of MFOs that confer pyrethroid resistance, which confirms previous work performed in the region by Ngufor et al. [14].

Aggregated data in the study area shows a higher SR indoors than outdoors in *An. gambiae* s.l., although no significant difference was observed. This might be due to the difference in the molecular species composition between indoor and outdoor. Indeed, the relative high ability of *An. gambiae* s.s. to get infected could have increased the SR indoors where it was in similar proportion with *An. coluzzii*, contrary to outdoors where its proportion was significantly lower. Similarly, indoor biting vectors were slightly older as shown by the parity rates observed. This is reminiscent of works by Machani et al. [37] who observed similar trend for SR in Bungoma and Kisian, Western Kenya. Although not significant, the SR was higher in *An. coluzzii* than in *An. gambiae* s.s. in the Alibori and Donga regions, Northern Benin [38]. The opposite trend observed in the present study might indicate the relative time in the age structure of the mosquito population, since sampling was performed only once.

The large variability in the EIR between villages confirms the heterogeneity in malaria transmission. Moreover, malaria transmission occurred mostly indoors than outdoors despite the use of conventional LLINs. This is in agreement with previous reports by Degefa et al. [39] in western Kenya, and might be due to the fact that most people slept indoors overnight and also the spread of resistance to pyrethroids incorporated in the conventional nets in use, as observed by Trape et al. [40] and, Ndiath et al. [41]. This emphasizes the need for assessing at the community level, the efficacy of dual-AI LLINs currently developed by the manufacturers, to control resistant mosquitoes.

Outdoor malaria transmission is a well-documented phenomenon in several other countries, including Cambodia [42], Peru [43], and Kenya [36]. According to Sherrard-Smith et al. [44], with a scenario of universal LLIN and IRS coverage (indoor control tools) across Africa, outdoor transmission could result in an estimated 10.6 million additional malaria cases over a year. In future, a major challenge to malaria control and elimination will likely be residual transmission as it remains a serious threat to the effectiveness of vector control tools that mostly target indoor malaria transmission.

Compared to *An. gambiae* s.l, *An. funestus* and *An. nili* likely play a minor role (lower EIR) in local malaria transmission due to their very low frequency (<2%), as observed in some Benin northern districts (Kerou and Pehunco) by Osse *et* al. [24]. However, the contribution of *An. funestus* and *An. nili* to malaria transmission was not evaluated in the present study. The seasonality and the single session of night sampling per village were also study limitations.

High resistance of malaria vectors to pyrethroids, as well as persistence of the disease transmission despite the strong culture of conventional LLINs use make the study area suitable for the implementation of the RCT that aims at assessing the efficacy of two dual-AI LLINs on malaria incidence, prevalence and transmission. For this assessment, baseline data gathered over the present study will serve for comparisons with the post-intervention ones.

## Conclusion

The present cross-sectional study provides information on key entomological indicators of malaria transmission prior to the implementation of the RCT. The mosquito relative

abundance shows that *An. gambiae* s.l. was the primary malaria vector followed by *An. funestus* and *An. nili*. *An. gambiae* s.l. was both highly resistant to pyrethroids and displayed multiple resistance mechanisms, L1014F *kdr* mutation and mixed function oxidases. HBR and EIR were higher indoors than outdoors in *An. gambiae* s.l. despite the high usage of conventional LLINs. This stresses the need for evaluating novel types of dual-AI LLINs that could help to better tackle malaria transmitted by pyrethroid resistant vectors.

## Supporting information

**S1 Fig. *An. gambiae* s.l. hourly biting rates in Cove, Zangnanado and Ouinhi.**
(DOCX)

**S1 Table. Parous rate in *An. gambiae* s.l. in Cove, Ouinhi and Zangnanando.**
(DOCX)

**S2 Table. Allelic frequencies of the L1014F *kdr* mutation in *An. gambiae* s.s and *An. coluzzii* collected using HLCs.**
(DOCX)

## Acknowledgments

We acknowledge the populations of the Cove, Ouinhi and Zangnanando districts as well as the local authorities who facilitated the implementation of the present study, through their close collaboration. The technicians who conducted mosquito processing and bioassays are thanked for their dedicated work, so is the LSHTM ODK support team that provided Electronic data solutions through LSHTM Open Research Kits (http://odk.lshtm.ac.uk/).

## Author Contributions

**Conceptualization:** Boulais Yovogan, Arthur Sovi, Constantin J. Adoha.

**Data curation:** Boulais Yovogan, Constantin J. Adoha, Edouard Dangbénon.

**Formal analysis:** Bruno Akinro, Saïd Chitou, Jackie Cook, Natacha Protopopoff.

**Funding acquisition:** Jackie Cook, Corine Ngufor, Mark Rowland, Natacha Protopopoff.

**Investigation:** Boulais Yovogan, Arthur Sovi, Constantin J. Adoha.

**Methodology:** Boulais Yovogan, Arthur Sovi, Constantin J. Adoha.

**Project administration:** Gil G. Padonou, Corine Ngufor, Martin C. Akogbéto.

**Software:** Edouard Dangbénon.

**Supervision:** Boulais Yovogan, Arthur Sovi, Gil G. Padonou, Constantin J. Adoha, Martin C. Akogbéto.

**Validation:** Jackie Cook, Immo Kleinschmidt, Mark Rowland, Natacha Protopopoff, Martin C. Akogbéto.

**Writing – original draft:** Arthur Sovi, Martin C. Akogbéto.

**Writing – review & editing:** Boulais Yovogan, Gil G. Padonou, Constantin J. Adoha, Bruno Akinro, Saïd Chitou, Manfred Accrombessi, Edouard Dangbénon, Hilaire Akpovi, Louisa A. Messenger, Razaki Ossè, Aurore Ogouyemi Hounto, Jackie Cook, Immo Kleinschmidt, Corine Ngufor, Mark Rowland, Natacha Protopopoff.

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
