## [Decision Letter · Decision Letter 0]

26 Apr 2021

PONE-D-21-02133

Pre-intervention characteristics of the mosquito species in Benin in preparation for a Randomized Controlled Trial assessing the efficacy of dual active-ingredient long-lasting insecticidal nets for controlling insecticide-resistant malaria vectors

PLOS ONE

Dear Dr. Sovi,

Thank you for submitting your manuscript to PLOS ONE. After careful consideration, we feel that it has merit but does not fully meet PLOS ONE’s publication criteria as it currently stands. Therefore, we invite you to submit a revised version of the manuscript that addresses the points raised during the review process.

Please clarify error bars on Figure 3,4 and 5 (CI or SEM?) and add in figure legend.

We look forward to receiving your revised manuscript.

Kind regards,

John Vontas

Academic Editor

PLOS ONE

Journal Requirements:

3) PLOS requires an ORCID iD for the corresponding author in Editorial Manager on papers submitted after December 6th, 2016. Please ensure that you have an ORCID iD and that it is validated in Editorial Manager. To do this, go to ‘Update my Information’ (in the upper left-hand corner of the main menu), and click on the Fetch/Validate link next to the ORCID field. This will take you to the ORCID site and allow you to create a new iD or authenticate a pre-existing iD in Editorial Manager. Please see the following video for instructions on linking an ORCID iD to your Editorial Manager account: https://www.youtube.com/watch?v=_xcclfuvtxQ

4) In your Methods, please provide more details about the participants consent, please clarify whether all the participants have been thoroughly informed of the risks of this study. In addition, please provide more details about what medical care/post treatment care was available for the participants, for example any precautions taken to try and protect fieldworkers from developing disease after infection, such as offering them antimalarial medication before being exposed to the mosquitoes.

5) Your ethics statement should only appear in the Methods section of your manuscript. If your ethics statement is written in any section besides the Methods, please move it to the Methods section and delete it from any other section. Please ensure that your ethics statement is included in your manuscript, as the ethics statement entered into the online submission form will not be published alongside your manuscript.

Reviewers' comments:

Reviewer's Responses to Questions

**Comments to the Author**

1. Is the manuscript technically sound, and do the data support the conclusions?

Reviewer #1: Yes

2. Has the statistical analysis been performed appropriately and rigorously? 

Reviewer #1: Yes

3. Have the authors made all data underlying the findings in their manuscript fully available?

Reviewer #1: Yes

4. Is the manuscript presented in an intelligible fashion and written in standard English?

Reviewer #1: Yes

5. Review Comments to the Author

Reviewer #1: Reviewer's report

Title: Pre-intervention characteristics of the mosquito species in Benin in preparation for a Randomized Controlled Trial assessing the efficacy of dual active-ingredient long- lasting insecticidal nets for controlling insecticide-resistant malaria vectors

General comments

This study of Pre-intervention characteristics of the mosquito species in Benin in preparation for a Randomized Controlled Trial assessing the efficacy of dual active-ingredient long- lasting insecticidal nets for controlling insecticide-resistant malaria vectors represents an important topic related to the bionomics of malaria vectors and insecticide resistance.

To study the entomological indicators in malaria vectors, human landing catch method is used. Prior this, informed consent was obtained from each collector. Moreover, the study protocol has been approved by the Benin national ethics committee for health research.

I could recommend the current version of this paper for publication after the very minor additive data below.

Version: 1

Date: 14 April 2021

Reviewer number: 1

Minor revisions

• In 178 to 179, Indicate the geographical coordinates of each district (Covè, Ouinhi and, Zagnanando)

in terms of longitude and latitude.

• Indicate the North symbol/direction (N) of the compass on figure 1 legend (Map of study area)

• Line 177 to 188: Authors should align the font of the paragraph similar to the rest text.

• Authors could choose to underlined each paragraph titles or not. If so, then they need to do throughout the document.

• Line 242 (Data management and analysis): Include in the table, the formulas for indoor and outdoor rates respectively.

• HBR, SR and EIR in An. gambiae s.l. in Cove, Ouinhi and Zangnanado were reported. Have authors also stratified EIR data also according to the An. gambiae species (i.e EIR in An. coluzzii and An. gambiae s.s respectively)? It will be interesting if available to include in the manuscript.

• Lines 304/305; 383/384: Because, Culex and Aedes represent the major mosquito's populations from the study area, authors should also discuss the strategies to reduce their effect on human, in the perspective of future LLINs trial, given the noising effect of Culex, and Aedes effect on human through dengue disease as described in many other countries.

• Line 327/328: What the error bars mean on Figure 3? Is it confidence intervals (CI) or Standard error of means (SEM)? Please add in figure legend.

• Line 367/368: What the error bars mean on Figure 4? Is it confidence intervals (CI) or Standard error of means (SEM)? Please add in figure legend.

• Line 369/370: What the error bars mean on Figure 5? Is it confidence intervals (CI) or Standard error of means (SEM)? Please add in figure legend

• Line 580 to 581: Please check for line spacing and homogenise accordingly

Level of interest: Statistical analysis added using Stata

Level of interest: An article of importance in its field

Quality of written English: Acceptable

6. PLOS authors have the option to publish the peer review history of their article (what does this mean?). If published, this will include your full peer review and any attached files.

Reviewer #1: No

---

## [Author Response · Author response to Decision Letter 0]

30 Apr 2021

All authors have agreed to changes made according to the comments from the reviewer and the editor.

Reviewer 1

1- In 178 to 179, Indicate the geographical coordinates of each district (Covè, Ouinhi and, Zagnanando) in terms of longitude and latitude.

Response: Done. See lines 178 & 179

2- Indicate the North symbol/direction (N) of the compass on figure 1 legend (Map of study area)

Response: Done. See the legend on figure 1

3- Line 177 to 188: Authors should align the font of the paragraph similar to the rest text.

Response: Corrected

4-Authors could choose to underlined each paragraph titles or not. If so, then they need to do throughout the document.

Response: We have now chosen not to underline anymore the paragraph titles and have been consistent throughout the test.

5- Line 242 (Data management and analysis): Include in the table, the formulas for indoor and outdoor rates respectively.

Response: For all calculated entomological indicators, the formulas used, were the same for both indoors and outdoors (See lines 259 & 260).

6- HBR, SR and EIR in An. gambiae s.l. in Cove, Ouinhi and Zangnanado were reported. Have authors also stratified EIR data also according to the An. gambiae species (i.e EIR in An. coluzzii and An. gambiae s.s. respectively)? It will be interesting if available to include in the manuscript.

Response: We agree with you that it would be interesting to include the EIR of An. gambiae s.s. and An. coluzzii in the manuscript, however we think that it may not show the real trend in terms of contribution of each molecular species to malaria transmission in the study area, given that only a sub-sample of collected specimens of An. gambiae s.l. have been tested through PCR for molecular species identification. 

7- Lines 304/305; 383/384: Because, Culex and Aedes represent the major mosquito's populations from the study area, authors should also discuss the strategies to reduce their effect on human, in the perspective of future LLINs trial, given the noising effect of Culex, and Aedes effect on human through dengue disease as described in many other countries.

Response: This is now discussed (See the discussion, from line 400 to line 406)

8- Line 327/328: What the error bars mean on Figure 3? Is it confidence intervals (CI) or Standard error of means (SEM)? Please add in figure legend.

Response: The error bars on Figure 3 are the confidence intervals, see lines 341/342.

9-Line 367/368: What the error bars mean on Figure 4? Is it confidence intervals (CI) or Standard error of means (SEM)? Please add in figure legend.

Response: The error bars on Figure 4 are the confidence intervals, see lines 381/382.

10-Line 369/370: What the error bars mean on Figure 5? Is it confidence intervals (CI) or Standard error of means (SEM)? Please add in figure legend

Response: The error bars on Figure 5 are the confidence intervals, see lines 383/384.

11-Line 580 to 581: Please check for line spacing and homogenise accordingly

Response: Corrected

Journal Requirements:

1-Please review your reference list to ensure that it is complete and correct. If you have cited papers that have been retracted, please include the rationale for doing so in the manuscript text, or remove these references and replace them with relevant current references. Any changes to the reference list should be mentioned in the rebuttal letter that accompanies your revised manuscript. If you need to cite a retracted article, indicate the article’s retracted status in the References list and also include a citation and full reference for the retraction notice.

Response: The reference list is complete and correct. No retracted article has been cited. 

2-Please ensure that your manuscript meets PLOS ONE's style requirements, including those for file naming

Response: The manuscript fully meets PLOS ONE's style requirements

3-PLOS requires an ORCID iD for the corresponding author in Editorial Manager on papers submitted after December 6th, 2016. Please ensure that you have an ORCID iD and that it is validated in Editorial Manager. To do this, go to ‘Update my Information’ (in the upper left-hand corner of the main menu), and click on the Fetch/Validate link next to the ORCID field. This will take you to the ORCID site and allow you to create a new iD or authenticate a pre-existing iD in Editorial Manager. Please see the following video for instructions on linking an ORCID iD to your Editorial Manager account: https://www.youtube.com/watch?v=_xcclfuvtxQ

Response: Done

4-In your Methods, please provide more details about the participants consent, please clarify whether all the participants have been thoroughly informed of the risks of this study. In addition, please provide more details about what medical care/post treatment care was available for the participants, for example any precautions taken to try and protect fieldworkers from developing disease after infection, such as offering them antimalarial medication before being exposed to the mosquitoes.

Response: See the ethical consideration section (From line 244 to 253)

5-Your ethics statement should only appear in the Methods section of your manuscript. If your ethics statement is written in any section besides the Methods, please move it to the Methods section and delete it from any other section. Please ensure that your ethics statement is included in your manuscript, as the ethics statement entered into the online submission form will not be published alongside your manuscript.

Response: Done 

We think that we have taken into account the recommendations and hope that the manuscript will be published soon.

Many thanks

---

## [Editor Report · Decision Letter 1]

3 May 2021

Pre-intervention characteristics of the mosquito species in Benin in preparation for a Randomized Controlled Trial assessing the efficacy of dual active-ingredient long-lasting insecticidal nets for controlling insecticide-resistant malaria vectors

PONE-D-21-02133R1

Dear Dr. ARTHUR SOVI,

We’re pleased to inform you that your manuscript has been judged scientifically suitable for publication and will be formally accepted for publication once it meets all outstanding technical requirements.

Kind regards,

John Vontas

Academic Editor

PLOS ONE
---

## [Editor Report · Acceptance letter]

12 May 2021

PONE-D-21-02133R1 

Pre-intervention characteristics of the mosquito species in Benin in preparation for a Randomized Controlled Trial assessing the efficacy of dual active-ingredient long-lasting insecticidal nets for controlling insecticide-resistant malaria vectors 

Dear Dr. Sovi:

I'm pleased to inform you that your manuscript has been deemed suitable for publication in PLOS ONE. Congratulations! Your manuscript is now with our production department. 

Kind regards, 

on behalf of

Prof. John Vontas 

Academic Editor

PLOS ONE